# Effect of Water and Ethanol Extracts from *Hericium erinaceus* Solid-State Fermented Wheat Product on the Protection and Repair of Brain Cells in Zebrafish Embryos

**DOI:** 10.3390/molecules26113297

**Published:** 2021-05-30

**Authors:** Shun-Kuo Sun, Chun-Yi Ho, Wei-Yang Yen, Su-Der Chen

**Affiliations:** Department of Food Science, National Ilan University, Number 1, Section 1, Shen-Lung Road, Yilan City 260007, Taiwan; sun3929@gmail.com (S.-K.S.); linca0730@gmail.com (C.-Y.H.); b9832107@ems.niu.edu.tw (W.-Y.Y.)

**Keywords:** *Hericium erinaceus*, neuroprotection, solid-state fermentation, zebrafish

## Abstract

Extracts from *Hericium erinaceus* can cause neural cells to produce nerve growth factor (NGF) and protect against neuron death. The objective of this study was to evaluate the effects of ethanol and hot water extracts from *H. erinaceus* solid-state fermented wheat product on the brain cells of zebrafish embryos in both pre-dosing protection mode and post-dosing repair mode. The results showed that 1% ethanol could effectively promote zebrafish embryo brain cell death. Both 200 ppm of ethanol and water extracts from *H. erinaceus* solid-state fermented wheat product protected brain cells and significantly reduced the death of brain cells caused by 1% ethanol treatment in zebrafish. Moreover, the zebrafish embryos were immersed in 1% ethanol for 4 h to cause brain cell damage and were then transferred and soaked in the 200 ppm of ethanol and water extracts from *H. erinaceus* solid-state fermented wheat product to restore the brain cells damaged by the 1% ethanol. However, the 200 ppm extracts from the unfermented wheat medium had no protective and repairing effects. Moreover, 200 ppm of ethanol and water extracts from *H. erinaceus* fruiting body had less significant protective and restorative effects on the brain cells of zebrafish embryos. Both the ethanol and hot water extracts from *H. erinaceus* solid-state fermented wheat product could protect and repair the brain cells of zebrafish embryos damaged by 1% ethanol. Therefore, it has great potential as a raw material for neuroprotective health product.

## 1. Introduction

*Hericium erinaceus* is a fungus commonly used in the East Asian diet. The bioactive components, such as erinacines and ericenones, in *Hericium erinaceus* have been widely discussed in neural studies. Ericenones was found earlier in *Hericium* fruiting body, including ericenones A-J [1,2,3,4]. Ericenones could promote the secretion of nerve growth factor (NGF) in a rat adrenal medulla pheochoursomocytoma cell line (PC12). The erinacines were primarily separated from the *Hericium* mycelium fermented liquid, including erinacines A-K [5,6,7,8] and erinacines P-R [9,10,11,12,13], and also had an effect in promoting the expression of NGF and the differentiation of axons in PC12 cells. Erinacines could induce relatively more NGF secretion than ericenones [7]. The aqueous polysaccharide extract of *Hericium erinaceus* could promote peripheral nerve regeneration after a nerve injury crush in rats [14]. The crude extracts of *Hericium erinaceus* mycelium inhibit purinoceptors to relieve neuropathic pain and neuro inflammation in mice [15].

However, in most of the animal and human experiments, *Hericium erinaceus* fruiting body and its extracts were used as the study object. They were not only supported by many cell experiments, but the protective effect in mice ischemic brain injury was also shown in animal models [16] to ameliorate learning and memory impairment in mice [17], rat sport repair neural damage [18], and human mild cognitive impairment [19]. An intervention of three 350 mg/g *H. erinaceus* mycelia enriched with 5 mg/g erinacine A capsules for 49 weeks achieved a better contrast sensitivity in mild Alzheimer’s disease patients, when compared to a placebo group, and may be important in achieving neurocognitive benefits [20].

Zebrafish (*Danio rerio*) belong to the carp species and are also known by the trade name zebra danio; they have a lifespan of about three and half years. They have often been used for vertebrate animal models in recent years. Because zebrafish showed a close similarity to and were homologous with other vertebrates in terms of molecules, biochemistry level, and cellular and physiological characteristics, they are often used as the study subject in molecular biology, developmental biology research, neurology, and genetics [21,22,23,24]. Zebrafish have numerous advantages in these studies, including that the embryos are transparent and suitable for observation in developmental and neurological studies. Moreover, zebrafish have a fast embryonic development and large quantity of offspring production, because they can spawn up to 70–300 eggs in one lot, and are very useful in genetics research [25].

In neurotoxicity studies, it was indicated that ethanol could affect learning and memory abilities in humans and rats [26,27,28,29]. In a study using a zebrafish model, it was shown that ethanol could cause deformities in the craniofacial skeleton, heart, and spinal cord of embryos and growth retardation [30,31,32,33]. The memory capacity also showed a significant reduction in the case of ethanol damage [34]. The zebrafish can be used as a good substitute animal model in assessing neurotoxicity due to the possibility of observing the number of its dead cells using staining techniques [35].

The objective of this study was to explore the protective and restorative effects of an ethanol extract (HEEE) and water extract (HEWE) from *Hericium erinaceus* solid-state fermented wheat product and ethanol extract (FBEE) and water extract (FBWE) from *Hericium erinaceus* fruiting body after ethanol damage to the brain cells of zebrafish embryos.

## 2. Results

### 2.1. Effects of the Different Ethanol Concentrations on the Death of Brain Cells in Zebrafish Embryos

The extent of the damage to the neural cells of zebrafish due to exposure to different concentrations of ethanol were investigated using 24 hpf zebrafish embryos immersed in different concentrations of ethanol (0, 0.2, 0.4, 0.8, and 1%) and treated for 24 h, followed by fluorescent staining. The fluorescent staining results showed that the number of neural cell deaths increased with the escalation of the ethanol concentration (Figure 1). Importantly, 1% ethanol resulted in more significant damage to the neural cells, but the 1% ethanol treatment did not kill the zebrafish embryos.

The fluorescent spots in the brain of the zebrafish embryos were counted and statistically analyzed (Figure 2). It was found that there was an average of 14 fluorescent spots with the 0.4% ethanol treatment, which was relatively higher than the 11 spots in the control group, and there was an increasing trend in fluorescent spots with the escalation of the ethanol concentration. When the concentration of ethanol reached 0.8%, the average number of fluorescence spots was 19, with significant differences from the control group; and the 1% ethanol treatment resulted in 29 fluorescent spots, which was a significantly increase and about 2.6-times the number of dead cells, compared to the control group. Therefore, 1% concentration of ethanol was selected for subsequent neural damage treatments.

### 2.2. The Effects of Different Concentrations of HEEE and HEWE Pretreatments on the Neural Cells in the Brain of Zebrafish Embryos

The mycelium and bioactive compound contents were changed during *Hericium erinaceus* solid-state fermentation, using wheat as the medium (Table 1). The mycelial was extracted with 99% methanol and then analyzed using HPLC with an ergosterol standard curve. The mycelium increased with the fermentation time. The crude polysaccharide in the *Hericium erinaceus* solid-state fermented wheat product was extracted with hot water (1:20) in a 300 W microwave for 5 min, and then the supernatant was added four times with ethanol to obtain the precipitate, which was analyzed according to the method of Doubis et al. [36]. In addition, the crude triterpenoid in the *Hericium erinaceus* solid-state fermented wheat product was extracted with 95% ethanol (1:20) in a 300 W microwave for 5 min, and the supernatant was analyzed using a spectrophotometer at 548 nm with an oleanolic acid standard curve [37]. The erinacine A was extracted with 85% ethanol (1:50) in a 400 W microwave for 7 min and then analyzed using HPLC [10]. Erinacine A was the secondary metabolite, which increased with the fermentation time. Both the highest crude polysaccharide and crude triterpenoid content appeared in the 35-day solid-state fermented wheat product; therefore, the *Hericium erinaceus* 35-day solid-state fermented wheat product was used in the following experiments for water and ethanol extraction.

Further investigation of the results of pretreatments with different concentrations of ethanol extract (HEEE) in zebrafish brain cells revealed protective effects. It was found that the zebrafish embryos that were only submerged in 1% ethanol for 24 h, but not treated with ethanol and water extracts (HEEE, HEWE), had significantly more fluorescently stained spots in the brain, compared to the quantized control group (Figure 3 and Figure 4). However, in the zebrafish embryos treated with HEEE and HEWE for 24 h, followed by the treatment with 1% ethanol, the number of apoptotic cells in the brain followed a downward trend, along with an increase in the HEEE and HEWE concentration. Among the groups, 200 ppm of the extract could maintain the minimum number of brain neural cell deaths and had no significant differences, compared with the control group. Therefore, a concentration of 200 ppm of HEEE and HEWE were used in the follow-up experiment.

### 2.3. The Effect of 200 ppm of the HEEE Post-Dosing Treatment in Repairing the Neural Cells in the Brain of Zebrafish Embryos

The 24 hpf zebrafish embryos were pre-soaked in 1% ethanol for different periods of time (1, 2, 4, 8, and 12h) and then transferred to 200 ppm HEEE to observe the growth and repairing situation of the zebrafish embryo brain cell damage at 48 hpf (Figure 5).

The results showed that, after the damage caused by 1% ethanol for different time periods, the number of zebrafish brain neural cell deaths varies with the immersion time and shows a significant increase. Moreover, in the groups of ethanol treatments for 1–4 h, 200 ppm of HEEE could significantly repair the brain cell damage and reduce the number of cell deaths. However, when the ethanol treatment time was extended to 8 and 12 h, HEEE could not effectively repair the neural cell damage. This is presumably because, when the time of the ethanol treatment increased, the time of the HEEE treatment was reduced, while the repairing effect was relatively reduced. Therefore, the ethanol treatment for 4 h was used in the following repair experiment to damage neural cells.

### 2.4. The Pre-Dosing and Post-Dosing Effects of Treatments with 200 ppm of 35-Day HEEE, 35-Day HWWE, 0-Day HEEE, 0-Day HEWE FBEE, and FBWE on the Neural Cells of Zebrafish Embryos

In this study, the zebrafish embryos were treated with ethanol and water extracts of *Hericium erinaceus* solid-state fermented wheat product, wheat medium, and *Hericium erinaceus* fruiting body in one of the two modes of pre-dosing or post-dosing ethanol damage (Figure 6 and Figure 7). It was found that, both in the pre-dosing and post-dosing mode, the ethanol and water extract of *Hericium erinaceus* solid-state fermented wheat product (35d HEEE, 35d HEWE) could both effectively protect and repair neural cells. Compared to the group with 1% alcohol damage alone, adding HEEE reduced the ratio of brain cell deaths to about 60%, and adding HEWE reduced it to about 80%. Moreover, both HEEE and HEWE could repair the brain cell damage and return it to the same level as the control group. The ethanol and water extracts of *Hericium erinaceus* fruiting body (FBEE and FBWE) were effective, but the degree of their protection and restoration were relatively poor. FBEE reduced the mortality ratio to about 20% in protecting the neural cells, and the repair degree was about 50%, while FBWE reduced the neural cell death rate to about 50% in protection mode, and the repair degree was about 60%. There was no effect in the ethanol and water extracts of unfermented wheat medium (0 d HEEE, 0 d HEWE) group, and the result showed there were no significant differences from the 1% ethanol damage group. This indicated that the extracts that protected and repaired the ethanol damage in the neural cells were not from wheat but rather from the *Hericium erinaceus* fermented wheat product.

Extract for 24 h → 1% Ethanol for 24 h → Image analyses

Water for 24 h →1% Ethanol for 4 h → Extract for 20 h → Image analyses

## 3. Discussion

### 3.1. Effects of Different Ethanol Concentrations on Brain Cell Deaths in Zebrafish Embryos

Carvan et al. [34] demonstrated that 300 mM (approximately 1.3%) ethanol significantly increased the number of brain neural cell deaths in zebrafish, and the results in this study showed the same trend. The zebrafish embryos were immersed in different concentrations of ethanol, AO fluorescent staining was employed, and the number of bright spots was counted. As shown in the study of Reimers et al. [32], 300 mM of ethanol did not significantly cause cell death in zebrafish embryos. As shown in other studies, the brain neural cells were damaged, and neural cell proliferation was reduced in the zebrafish embryos damaged by ethanol [35].

### 3.2. The Effects of Different Concentrations of HEEE and HEWE Pretreatments on the Neural Cells in the Brain of Zebrafish Embryos

The polysaccharide extracted from astragalus reduced apoptosis by inducing TERT and MEM2 protein expression and reducing the apoptotic pathway protein expression, such as Bax and p53, in a zebrafish model [38]. Cell apoptosis was reduced, and the growth and development of the individual was inhibited, which could have an anti-aging effect on adult individuals.

As for the neural protective effect, ethanol extracts from some Chinese medicines, such as *Eriocaulon buergerianum* and *Alpinia oxyphylla*, had a significant effect on dopamine neuroprotection in zebrafish [39,40]. The protection mechanism showed that these two permeable extracts had anti-inflammatory and antioxidant effects, reduced the mRNA expression of IL-1b and TNF-α and other inflammatory cytokines, and reduced the production of NO and iNOS. Moreover, a protective effect on the neurons was achieved. Rosemberg et al. [41]. showed similar results in a study on taurine. Taurine pretreatment reduced the amount of thiobarbituric acid reactive substances (TBARS) in zebrafish brains damaged by 1% ethanol. Moreover, it increased the superoxide dismutase (SOD) content and reduced the alcohol damage in the brain by improving the antioxidant capacity. The treatment with a dry extract of *Gynostemma pentaphyllum* (var. Ginpent), which is rich in secondary metabolites with potential antioxidant and anti-inflammatory properties, induced a recovery of motor function, measured with the rotarod test and pole test [42]. Other studies also showed that antioxidants reduced the damage caused by ethanol in zebrafish and mice [43,44,45]. The freeze-dried broth of *Hericium erinaceus* fermented liquid products contained 2.75 mg/g of total phenolics and 0.25 mg/g of flavonoids. It also had high antioxidant activities, such as a DPPH scavenging ability up to 67.24% and ferrous ion chelating ability of 85.41% [46]. This indicates that *Hericium erinaceus* fermented products have a good antioxidant capacity.

Mitchell et al. [47] demonstrated that, besides the antioxidant and anti-inflammatory factors, nerve growth factor (NGF) and brain-derived neurotrophic factor (BDNF) effectively protected the hippocampal gyrus cells, reducing the cell death caused by ethanol damage in vitro experiments. Shimbo et al. [48] fed three-week-old rat pups daily with 8 mg/kg of erinacine A for 14 consecutive days and detected the brain tissue NGF content. It was found that the NGF content showed a relatively significant increase in the locus coeruleus and hippocampus portions, compared to the control group. A notably higher content of about 12 times the NGF was found in the locus coeruleus. In the study of Hazekawa et al. [16], the increased NGF in the brain tissue protected from subsequent neural damage caused by the brain ischemia. The ethanol extracts of *Hericium erinaceus* solid-state fermented products could induce NGF expression in PC12 cells, with an NGF content of up to five times that of the control group, which is also significantly higher than that of the ethanol extract of *Hericium erinaceus* fruiting body [49].

### 3.3. The Effect of 200 ppm of the HEEE Post-Dosing Treatment in Repairing the Neural Cells in the Brain of Zebrafish Embryos

In rodent studies, Tapia-Arancibia et al [50]. Found a significantly decreased mRNA expression of BDNF in the CA1 and dentate gyrus of the hippocampus and the visual part of the nucleus of the hypothalamus in ethanol-treated rats. This was due to the ethanol interference N-methyl-D-aspartate (NMDA) receptor, which inhibited the BDNF expression [51]. The NGF content in the cerebral cortex and hippocampus also dropped significantly in the ethanol-treated rats. It is noteworthy that the secretion of BDNF and NGF may affect the content of neurotransmitters, and the growth of neural axons and survival of neurons are closely related to it [49,52,53]. This led to the loss of brain neurons after ethanol damage [54]. However, it was found that the mRNA expression of BDNF was significantly increased in the cerebral cortex, hypothalamus nucleus, and hippocampus CA3 region, compared with the control group 12 h after stopping ethanol consumption [50]. This result may echo those of the 8-h and 12-h groups, which did not show a significant repairing effect on ethanol-damaged cells in this study. The changes in these neurotrophic factors are considered to perform a protective role in the repairing and regeneration of neural damage [55]. Phytoextracts of *Artocarpus lakoocha* Roxb. and *Artocarpus heterophyllus* Lam. flowers showed antioxidant activity and increased the expression of both Nrf2 (promoter gene) and HO-1 and NQO1 (Nrf2-regulated genes) and actively modulated the genes involved in the expression of the redox enzymatic pathways [56].

The development of the zebrafish brain showed that there was a relatively large number of growth of axons in the embryos between the 1 dpf and 2 dpf stages [57]. This observation was coherent with the study of Hashimoto and Heinrich [58]. Zebrafish embryos in different developmental stages were investigated, and the results show the amount of BDNF mRNA correlated to the developmental stage. The BDNF mRNA expression during embryonic development at 1 dpf is low and increases significantly after 1 dpf. The BDNF mRNA expression at 3 dpf is six times as high as that at 1 dpf. This indicates that BDNF may also play an important role in the neural system development of zebrafish. Weis [59] confirmed that the administration of exogenous NGF increased the size and number of zebrafish spinal neural junctions. Therefore, the results of this study indicate that ethanol damage could cause neural cell death and prevent growth by reducing the expression of neurotrophic factors, which was highly expressed after 1 dpf to promote neural development. The ethanol extracts of *Hericium erinaceus* had an inducing effect on NGF expression in PC12 cells [49]. HEEE also helped zebrafish brain cells to produce neurotrophic factors when the ethanol damage was stopped and then repaired the damaged neurons.

### 3.4. The Pre-Dosing and Post-Dosing Effect of Treatments with 200 ppm of 35-Day HEEE, 35-Day HWWE, 0-Day HEEE, 0-Day HEWE FBEE, and FBWE on the Neural cells of Zebrafish Embryos

In the early study of Kawagishi et al. [2], hericenones isolated from fruiting body of *Hericium erinaceus* extract could promote neural cell growth. In recent years, Hazekawa et al. [16] and Mori et al. [17] also confirmed the neural protective and memory-improving effect of *Hericium erinaceus* in mice. Wong et al. [14] found that polysaccharide extract of *Hericium erinaceus* fruiting body could promote peripheral nerve regeneration after a nerve injury crush in rats. The effect of *Hericium erinaceus* on neural repair was also indicated by Mori et al. [60] in a clinical experiment. Patients with a mild cognitive impairment taking *Hericium* capsules showed an improvement but then a decline in cognitive scores four weeks after discontinuing the *Hericium* treatment.

## 4. Materials and Methods

### 4.1. Materials

Wheat purchased from the Xingfa grain store (YiIan, Taiwan) was used to prepare the medium. *Hericium erinaceus* (BCRC 36470) was purchased from the Bioresource Collection and Research Center (Hsinchu, Taiwan). Further, 95% ethanol reagent grade, magnesium sulfate (MgSO_4_‧7H_2_O), dipotassium hydrogen phosphate (K_2_HPO_4_), and glucose (C_6_H_12_O_6_) were purchased from Wako Pure Chemical Industries, Ltd. (Osaka, Japan). Yeast extract, potato dextrose agar medium (PDA), and potato dextrose broth (PDB) were purchased from Difco Co. (Sparks, MD, USA). Tween 20 and phosphate buffer solution (PBS) were purchased from Gibco (Grand Island, NY, USA). Additionally, 99% reagent grade ethanol was purchased from Shimakyu Chemical Co., Ltd. (Japan). Acridine orange (AO) was purchased from Sigma (St. Louis, MO, USA). Wild-type zebrafish (AS-AB) was purchased from the Taiwan zebrafish center branch (TZCAS, Taipei, Taiwan) in Academia Sinica.

### 4.2. Equipment

The 12-microplate (Cat. 3915) was purchased from BD (FALCON, USA). The aquatic farming systems were purchased from Bio Promotion Co. Ltd. (New Taipei, Taiwan). Inverted fluorescence microscope (Olympus 1 × 71). The following equipment was used: an ultrasonic cleaner, DC-600H (DELTA, New Taipei, Taiwan); a vertical pressure steam autoclave (TM-329, Tomin medical equipment Co., Ltd., USA); a concentrated under reduced pressure machine (Eyela Oil Bath Osb-2000, Tokyo, Japan); a 25 °C shaking incubator (YIH DER, LM-600R, USA); a 28°C shaking incubator (YIH DER, LM-600R, USA). The stereo microscope (HG208194) was purchased from Sun Optics. The microwave extraction device in our laboratory was self-assembled, including an adjustable microwave power microwave (Bio Promotion Co. Ltd., New Taipei, Taiwan), consisting of a microwave oven with a flat top and a flask condenser with a volume of 500 mL. Then, a 4°C cold water cycle machine (Firstek, 8402H, New Taipei, Taiwan) was added to maintain the condenser at a low temperature.

### 4.3. Cultivation of Hericium erinaceus

#### 4.3.1. Activation of *Hericium erinaceus*

*Hericium erinaceus* (BCRC 36470) was placed on a PDA plate in the 25 °C incubator for monthly sub-culturing. Moreover, the outgrowth mycelium was inoculated into a 500 mL flask with baffles containing 5% glucose broth (containing 5% glucose, 0.1% yeast extract, 0.1% dipotassium hydrogen phosphate, 0.05% magnesium sulfate, and quantitatively distilled water to 150 mL) and pre-activated for seven days at 25 °C under a 150 rpm shaking condition.

#### 4.3.2. Cultivation of *Hericium erinaceus* Solid-State Fermented Wheat Product

The pre-activated *Hericium erinaceus* broth was used to inoculate a 10 mL sterilized 500 g wheat medium, which was cultured in an incubator at 25 °C. The fermented wheat product was sterilized for 60 min at 121 °C in an autoclave and was ground to powder after 50 °C hot air drying. Finally, the 0-, 21-, 28-, 35-, 42-, and 56-day fermented products were analyzed for mycelium, crude polysaccharide, crude triterpenoids, and erinacine A [37]. The ethanol and hot water were extracted from *Hericium erinaceus s*olid-state fermented wheat product and fruiting body

### 4.4. Extraction of Hericium erinaceus Solid-State Fermented Wheat Product

The *Hericium erinaceus* 35-day solid-state fermented wheat product, wheat medium, and *Hericium* fruiting body were extracted with 95% ethanol and distilled water in the microwave extraction conditions, with a solid–liquid ratio of 1:20 and 5 min of extraction under 300 W, and the extract was centrifuged at 6000 rpm for 15 min. The supernatant of the extract was vacuum evaporated until dry. The ethanol (HEEE) and hot water extracts (HEWE) from *Hericium erinaceus* 35-day solid-state fermented wheat product and the ethanol (FBEE) and hot water (FBWE) extracts from *Hericium erinaceus* fruiting body were re-suspended with distilled water under ultrasonic shock and kept for later use.

### 4.5. The Breeding and Fertilized Zebrafish Egg Collection 

Wild-type zebrafish were reared in a water fertility system, and male and female fish were placed in separated tanks. The water temperature was maintained at 28 ± 1 °C, and the photo period was used to domesticate the zebrafish’s natural spawning. The light was adjusted to 14 h of light and 10 h of dark. The day before harvesting the fertilized egg, the male fish and female fish were placed in the same small breeding tanks before turning the lights out, and marbles were placed on the bottom of the tank to collect the fertilized eggs laid as a consequence of natural mating the next morning.

### 4.6. Dosing Culture of Zebrafish Embryos

The dissolution of the sample was adjusted to the desired concentration and disposed into a plate with a diameter of 10 cm. All treatments were 20 mL in volume.

#### 4.6.1. Pre-dosing Treatment

After the fertilized eggs were collected, impurities and bad unfertilized eggs were immediately removed and placed in the plate in different concentrations (10, 100, and 200 ppm) of ethanol extracts (HEEE, FBEE) or hot water extracts (HEWE, FBWE). The control group was placed in distilled water and incubated in a 28 °C incubator for 24 h. After the removal of the extracts, the embryos were washed with PBST (×1 phosphate buffered solution PBS + 0.05% Tween 20) three times, soaked in 1% ethanol, and incubated in a 28 °C incubator for 24 h. After the treatment, the ethanol was removed, and the embryos were washed with PBST three times.

#### 4.6.2. Post-Dosing Treatment

After the fertilized eggs were collected, impurities and bad unfertilized eggs were immediately removed, and all the fertilized eggs were picked off, placed in distilled water, and incubated in a 28 °C incubator. After 24 h of incubation, the fertilized eggs were grouped and placed in 1% ethanol in a 28 °C incubator for 1, 2, 4, 8, and 12 h. The control group was placed in distilled water. At the end of treatment, the ethanol was removed, and the embryos were washed with PBST three times and soaked in 200 ppm of ethanol extract (HEEE). The treated embryos were cultured in a 28°C incubator until they reached the 48 hpf stage. After the end of the treatment and the removal of the extract, the embryos were washed three times by PBST (after confirming the time period of the damage caused by the 1% ethanol, and the HEWE, FBEE, FBWE, and hot water and ethanol extracts of the wheat medium were compared).

#### 4.6.3. Fluorescent Staining of Zebrafish Embryos

At the end of the drug treatment, the embryos were washed with PBST three times, soaked in 5 μg/mL AO fluorescent dyes, and stained for 60 min in the dark. After the staining, the embryos were washed with PBST three times for 5 min each. The stained zebrafish embryos were observed with a green fluorescent microscope containing an FITC filter (excitation: 488 nm, emission: 515 nm) and photographed for the records.

### 4.7. Statistical Analysis

The test results were presented as the mean ± standard deviation. The data obtained from the study were statistically analyzed using the Statistical Package for Social Science version 14.0 statistical software package (SPSS, SPSS INC., Wang Tak International Software Consultants Ltd.). A Duncan’s Multiple Range Test, with a significance level of α = 0.05, was used to compare the statistical significance of the differences.

## 5. Conclusions

Zebrafish embryos treated with 1% ethanol were used as a simple and effective neuroprotective platform for testing the protection and repair of brain cells. In short, 200 ppm of ethanol (HEEE) and hot water (HEWE) extracts from *Hericium erinaceus* solid-state fermented wheat product had significant protective and repairing effects on 1% ethanol damage to neural cells in zebrafish. Based on the protective and restorative neural effects, *Hericium erinaceu*s solid-state fermented wheat product has a greater potential as a raw material for neuroprotective functional food products than *Hericium erinaceu*s fruiting body.

## Figures and Tables

**Figure 1 molecules-26-03297-f001:**
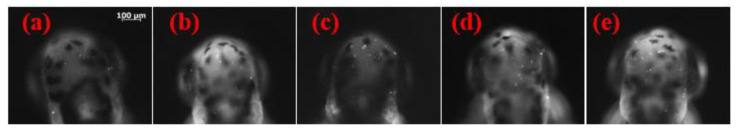
Effect of different ethanol concentrations on the brain cells of zebrafish embryos. Twenty-four dpf zebrafish were exposed to different concentrations of ethanol for 24 h: (**a**) control; (**b**) 0.2% ethanol; (**c**) 0.4% ethanol; (**d**) 0.8% ethanol; (**e**) 1% ethanol. Photomicrographs were taken from embryos at 48 hpf. Dorsal view, anterior to top.

**Figure 2 molecules-26-03297-f002:**
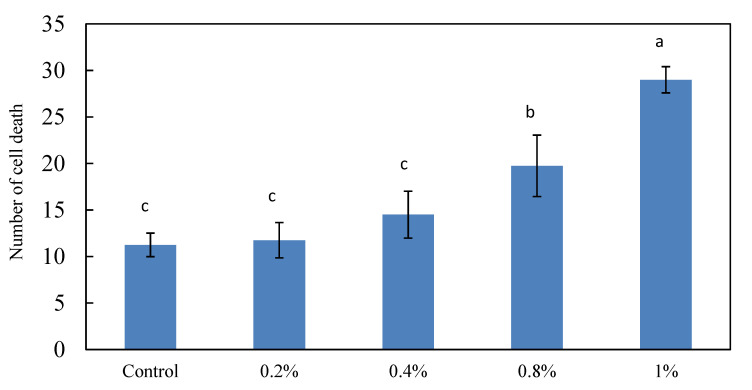
Effect of different ethanol concentrations on the brain cell death of zebrafish embryos. Twenty-four dpf zebrafish were exposed to different concentrations of ethanol for 24 h, except the control. The bar graphs represent the mean number of acridine orange-labeled cells, mean ± S.D.; n = 4 embryos. ^a–c^ The mean dead cell values with different letters are significantly different (*p* < 0.05).

**Figure 3 molecules-26-03297-f003:**
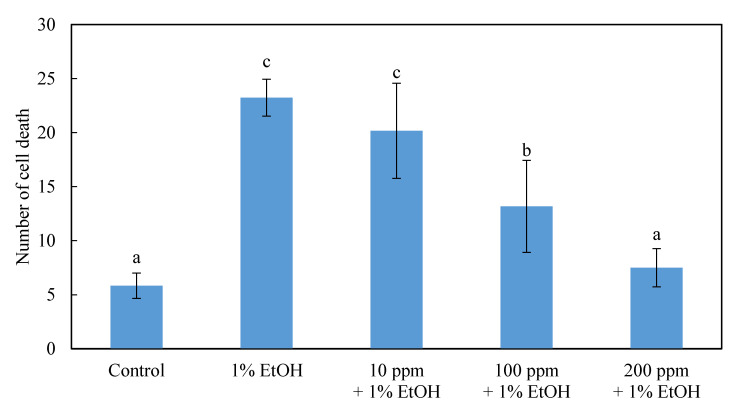
Effect of pretreatment with different concentrations of an ethanol extract of the *Hericium erinaceus* solid-state fermented wheat product (effect of HEEE on the brain cells of zebrafish embryo). The bar graphs represent the mean number of acridine orange-labeled cells, mean ± S.D.; *n* = 8 embryos. ^a–c^ The mean cell death values with different letters are significantly different (*p* < 0.05).

**Figure 4 molecules-26-03297-f004:**
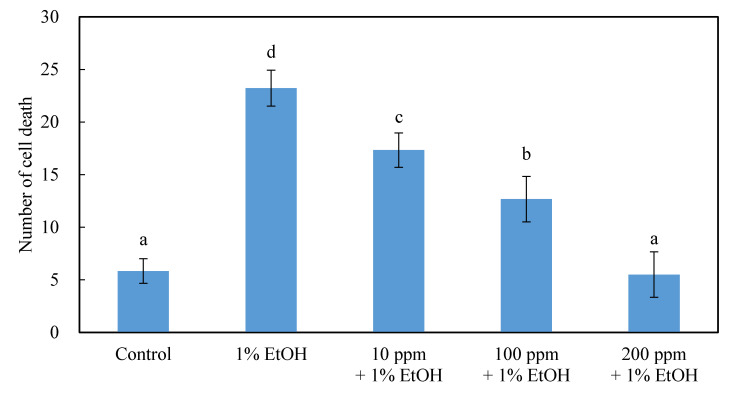
Effect of pretreatment with different concentrations of a hot water extract of *Hericium erinaceus* solid-state fermented wheat product (HEWE) on the brain cells of zebrafish embryo. The bar graphs represent the mean number of acridine orange-labeled cells, mean ± S.D.; *n* = 8 embryos. ^a–d^ The mean cell death values with different letters are significantly different (*p* < 0.05).

**Figure 5 molecules-26-03297-f005:**
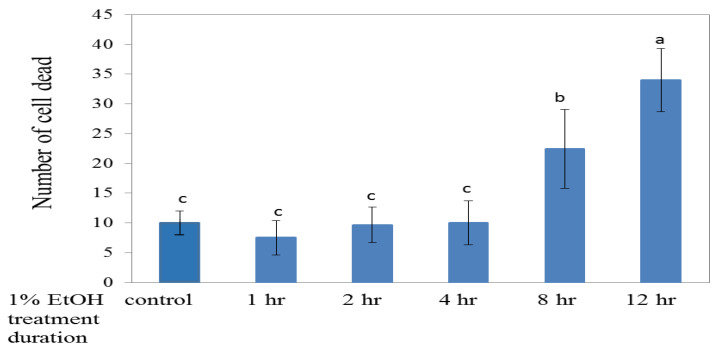
Effect of post-treatment with 200 ppm of HEEE on the cells of zebrafish embryos. The bar graphs represent the mean number of acridine orange-labeled cells, mean ± S.D.; *n* = 6 embryos. ^a–e^ The mean cell death values with different letters are significantly different for the different ethanol treatment times (*p* < 0.05). * The means of the cell death values with * are significantly different for the same ethanol treatment time (*p* < 0.05).

**Figure 6 molecules-26-03297-f006:**
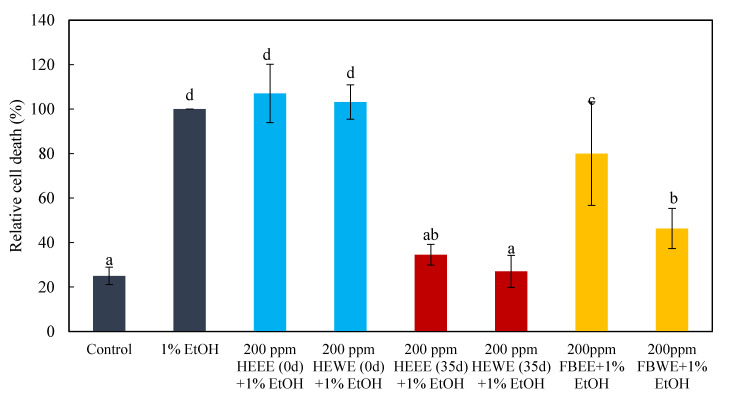
Effect of pretreatment with the hot water and ethanol extract of *Hericium erinaceus* solid-state fermented wheat product, fruiting body, and wheat on the brain cells of zebrafish embryos. The bar graphs represent the mean number of acridine orange-labeled cells, mean ± S.D.; *n* = 6 embryos. ^a–d^ The mean cell death values with different letters are significantly different (*p* < 0.05).

**Figure 7 molecules-26-03297-f007:**
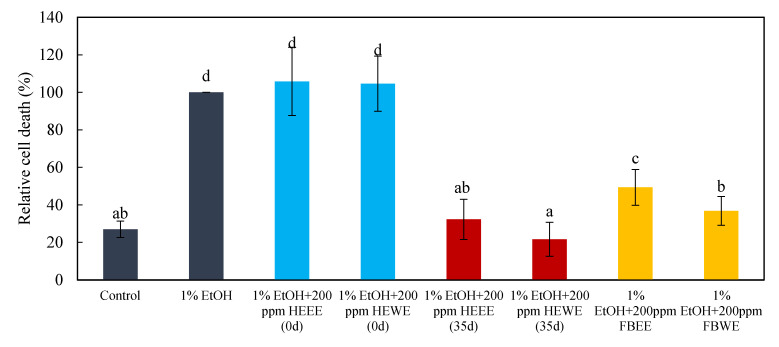
Effect of post-treatment with the hot water and ethanol extract of *Hericium erinaceus* solid-state fermented wheat product, fruiting body, and wheat on the brain cells of zebrafish embryos. The bar graphs represent the mean number of acridine orange-labeled cells, mean ± S.D.; *n* = 6 embryos. ^a–d^ The mean cell death values with different letters are significantly different (*p* < 0.05).

**Table 1 molecules-26-03297-t001:** Effect of fermentation time on mycelium and bioactive compound contents in *Hericium erinaceus* solid-state fermented wheat product.

Time (day)	Mycelium (%)	Crude Polysaccharide (%)	Crude Triterpenoid (%)	Erinacine A (%)
0	-	10.402 ± 0.392 ^d^	0.054 ± 0.025 ^d^	0.003 ± 0.002 ^c^
21	2.118 ± 0.015 ^d^	11.749 ± 0.321 ^d^	0.071 ± 0.027 ^c^	0.029 ± 0.002 ^b^
28	2.978 ± 0.065 ^d^	13.412 ± 0.691 ^b^	0.108 ± 0.027 ^b^	0.031 ± 0.002 ^b^
35	3.500 ± 0.044 ^c^	15.649 ± 0.931 ^a^	0.153 ± 0.029 ^a^	0.042 ± 0.002 ^b^
42	5.846 ± 0.200 ^b^	10.764 ± 0.315 ^d^	0.077 ± 0.029 ^c^	0.051 ± 0.002 ^ab^
56	7.415 ± 0.251 ^a^	10.323 ± 0.464 ^d^	0.071 ± 0.029 ^c^	0.062 ± 0.004 ^a^

^a–d^ Means in the same column with different superscript letters are significantly different (*p* < 0.05).

## Data Availability

The data presented in this study are available in this article.

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
