# Peer review of "Effect of Water and Ethanol Extracts from Hericium erinaceus Solid-State Fermented Wheat Product on the Protection and Repair of Brain Cells in Zebrafish Embryos"

_molecules, 2021, doi:10.3390/molecules26113297_

Round 1
Reviewer 1 Report
The paper presents the effects of the ethanol and hot water extracts from Hericium erinaceus solid-state fermented wheat product on brain cells of zebrafish embryo in both pre-dosing protection mode and post-dosing repair mode.
The work is of general interest, is well planned and described. In my opinion the paper is worth studying and the manuscript contains enough original material. The experimental tests are carried out correctly using appropriate methods. The results are quite interesting and well statistically analyzed.
Minor corrections:
Text formatting should be carefully checked.
The language should be modified carefully.
Author Response
The manuscript submitted to MDPI for English editing has been edited.

Reviewer 2 Report
The manuscript by Sun and colleagues is of particular interest. The authors use some extracts (in water and ethanol) of the products derived from fermented wheat from Hericium erinaceus to evaluate the protection from the toxic action of ethanol in Zebrafish cultures. The toxic effect of ethanol on cellular modeling is known, but especially during the growth phases of Zebrafish. These extracts appear to protect against ethanol toxicity with surprising results.
The manuscript is clear, although some typos should be corrected by the authors. I suggest to the authors these two interesting manuscripts which could enrich the references of their manuscript:
Gupta AK et al. Artocarpus lakoocha Roxb. and Artocarpus heterophyllus Lam. Flowers: New Sources of Bioactive Compounds. Plants (Basel). 2020
Mastinu A et al Protective Effects of Gynostemma pentaphyllum (var. Ginpent) against Lipopolysaccharide-Induced Inflammation and Motor Alteration in Mice. Molecules. 2021
Author Response
The manuscript submitted to MDPI for English editing has been edited.
The treatment with dry extract of Gynostemma pentaphyllum (var. Ginpent) which is rich in secondary metabolites with potential antioxidant and anti-inflammatory properties, induced a recovery of motor function measured with the rotarod test and pole test. [Mastina et al., 2021]
The phytoextracts of Artocarpus lakoocha Roxb. and Artocarpus heterophyllus Lam. flowers showed an antioxidant activity, and increased the expression of both Nrf2 (pro-moter gene) and HO-1 and NQO1 (Nrf2-regulated genes) and also actively modulated the genes involved in the expression of the redox enzymatic pathways [Gupta et al, 2020].

Reviewer 3 Report
The authors evaluated the effect of water and ethanol extracts of Hericium erinaceus fermented wheat product on ethanol-induced brain damage in zebrafish embryos.
The data is not enough for publication in Molecules. Or the theme doesn’t match to this journal even for the SI “Food Chemistry”.
Major comments:
- Hericium erinaceus (HE) itself already reported to improve several kinds of brain diseases. I cannot understand the advantage to use HE-fermented wheat products in this study.
- The ingredients of HEEE and HEWE are necessary.
- Mechanisms of action in HEEE and HEWE are also necessary.
Author Response
The manuscript submitted to MDPI for English editing has been edited.
1. The mycelium and bioactive compound contents were changed during Hericium er-inaceus solid-state fermentation using wheat as medium (Table 1). The mycelial was extracted by 99% methanol and then analyzed by HPLC with ergosterol standard curve; the mycelium increased with fermentation time.
2. The crude polysaccharide in the Hericium erinaceus solid-state fermented wheat products was extracted by hot water (1:20) using 300 W microwave for 5 min, and then the supernatant was added 4 fold ethanol to obtain precipitate, which were analyzed according to Doubis et al. method [36]. In addi-tion, the crude triterpenoid in the Hericium erinaceus solid-state fermented wheat prod-ucts was extracted by 95% ethanol (1:20) using 300 W microwave for 5 min, and the su-pernatant was analyzed by spectrophotometer at 548 nm with oleanolic acid standard curve [37]. The erinacine A was extracted by 85% ethanol (1:50) using 400 W microwave for 7 min, and then analyzed by HPLC [10].
3. Mitchell et al. [46] demonstrated that besides antioxidant and anti-inflammatory fac-tors, nerve growth factor (NGF) and brain-derived neurotrophic factor (BDNF) effectively protected the hippocampal gyrus cells, reducing cell death caused by ethanol damage us-ing in vitro experiments. Shimbo et al. [47] fed three-weeks old rat pups daily with 8 mg/kg of erinacine A for 14 consecutive days, and detected the brain tissue NGF content. It was found NGF content relatively significant increase in the locus coeruleus and hippo-campus portion compared to the control group. Particularly higher content of about 12 times the NGF was found in the locus coeruleus. In Hazekawa et al. [16] study, the in-creased NGF in the brain tissue protected a subsequent neural damage caused by brain ischemia. The ethanol extracts of Hericium erinaceus solid-state fermented products could induce NGF expression in PC12 cells, the NGF content up to five times to the control group, and also significantly higher than ethanol extract of Hericium erinaceus fruiting bodies [48].

Reviewer 4 Report
The paper "Effect of Water and Ethanol Extracts from Hericium erinaceus Solid-state Fermented Wheat Product on Protection and Repair of Brain Cells in Zebrafish Embryos" is a good work. In my opinion the paper can be accepted pending revisions, particularly:
- check the paper for the significant digit
- i suggest to deeper highlight the novelty. Into the text is not clear and often omissed
- clarify for the different assays the control used and the conditions
Author Response
The manuscript submitted to MDPI for English editing has been edited.
4.6.1. Pre-dosing treatment
After the fertilized eggs were collected, impurities and bad unfertilized eggs were immediately removed and were placed in the plate containing different concentrations (10, 100, and 200 ppm) of ethanol extracts (HEEE, FBEE) or hot water extracts (HEWE, FBWE). The control group was placed in distilled water, and incubated at 28°C incubator for 24 hours. After the removal of the extracts, embryos were washed with PBST (x1 phosphate buffered solution PBS + 0.05% Tween 20) three times, and then soaked in 1% ethanol and incubated at 28°C incubator for 24 hours. After the treatment, ethanol was removed, and embryos were washed with PBST three times.
4.6.2. Post-dosing treatment
After the fertilized eggs were collected, impurities and bad unfertilized eggs were immediately removed, and all the fertilized eggs were picked off and placed in distilled water, and incubated at 28°C incubator. After 24 hours incubation, the fertilized eggs were grouped and placed in 1% ethanol, 28℃ incubator for 1, 2, 4, 8, 12 hours, respectively. The control group was placed in distilled water. At the end of treatment, ethanol was removed, and the embryos were washed with PBST three times, and then soaked in 200 ppm of ethanol extract (HEEE). Treated embryos were cultured at 28°C incubator until embryos reached 48 hpf stage. After the end of treatment, the removal of the extract was followed by three times PBST wash for the embryos (after confirmed the damage time period of 1% ethanol, HEWE, FBEE, FBWE, and hot water and ethanol extracts of wheat medium were brought to comparison).

Round 2
Reviewer 3 Report
1&2. Table 1 is not enough for the list of bioative ingreduents. To insist the advantage of HE solid state fermented wheat, you have to compare it with HE alone.
3. It's just a speculation. Better to prove some of them as a scientific paper.
Reviewer 4 Report
The criticisms were revised in the current version that can be accepted for publication